# Challenges in Global Distribution and Equitable Access to Monkeypox Vaccines

**DOI:** 10.3390/v16121815

**Published:** 2024-11-21

**Authors:** Nengak P. Danladi, Progress Agboola, Peter Olaniyi, Solomon Eze, Oluwatimilehin Oladapo, Danielle Obiwulu, Olatokun Shamsudeen Akano, Olowoyeye Aishat Adeola, Khaliq Olawale, Azeez Idowu Adiatu, Agboola Peace

**Affiliations:** 1Global Health Infectious Diseases and Control Institute, Nasarawa State University Keffi, RWR4+H9P, Keffi 961101, Nigeria; 2African Community for Systematic Review and Meta-Analysis, 172 Akai Efa, MCC Road, Calabar 540211, Cross River State, Nigeria; 3Department of Medicine and Surgery, Ladoke Akintola University of Technology, Ogbomoso 210214, Nigeria; agboolaprogress@gmail.com (P.A.); peterolaniyi17@gmail.com (P.O.); 4Department of Biochemistry, Abia State University, Uturu 441103, Nigeria; chaplainsolomoneze@gmail.com; 5College of Medicine, University of Ibadan, Ibadan 200001, Nigeria; timmidapo@gmail.com (O.O.); olatokunshamsudeen@gmail.com (O.S.A.); 6College of Medicine, University of Lagos, Lagos 102216, Nigeria; danielleobiwulu@gmail.com; 7College of Health Sciences, Obafemi Awolowo University, Ile-Ife 220103, Nigeria; flowergeal56@gmail.com; 8Department of Medical Rehabilitation, College of Health Sciences, Obafemi Awolowo University, Ile-Ife 220103, Nigeria; olawalekhaliq@gmail.com; 9Faculty of Pharmacy, University of Ibadan, Ibadan 200001, Nigeria; adiatuazeez310@gmail.com; 10Seventh-Day Adventist College of Nursing, Ile-Ife 220103, Nigeria; peaceagboola60@gmail.com

**Keywords:** monkeypox, challenges, equitable, vaccines, distribution

## Abstract

The monkeypox outbreak has grown beyond the regions in which it was considered endemic. It has spread from central and west Africa to non-endemic regions like Europe, America, and other parts of the world. It has recently been classified as a public health emergency of international concern. This study evaluated the challenges faced globally and equitable access to monkeypox vaccines. Global competition has been observed in the race to obtain vaccines, with low- and middle-income countries being disadvantaged. Great inequity exists in the distribution of vaccines globally through advance purchase agreements, vaccine stockpiling, vaccine nationalism, the inequitable distribution of existing resources, and insufficient surveillance and reporting mechanisms. To address some of these challenges, there is a need for strengthening the global vaccine manufacturing capacity, targeting countries with elevated risk profiles and limited resources, strengthening surveillance systems, and addressing vaccine hesitancy.

## 1. Introduction

Caused by the MPOX virus (MPXV), monkeypox is a zoonotic double-stranded DNA virus belonging to the Orthopoxvirus genus within the Poxviridae family [1,2]. This viral family also includes the variola virus, the causative agent of smallpox, a disease eradicated over forty-four years ago following widespread immunization efforts using the closely related vaccinia virus (VACV) [3]. MPOX’s first human case was in 1970 in the Democratic Republic of Congo (DRC). Since then, multiple cases have been reported across the tropical rainforest regions of western and central Africa [4]. However, the MPOX virus (MPXV) outbreak has expanded beyond its traditional endemic areas of central and west Africa, spreading to multiple non-endemic countries across Europe, the Americas, and other parts of the world [2,4]. Five notable MPOX outbreaks have been documented, occurring in 1970, 1996–1997, 2003, and 2018 [3,5,6,7,8]. Between January 2022 and August 2024, more than 120 countries documented MPOX cases, with over 100,000 confirmed cases in laboratories and more than 220 fatalities reported among these confirmed cases [2,9].

In the USA, 71 individuals were infected in 2003, with laboratory confirmation in 35 cases, due to contact between prairie dogs and African rodents. From 2018 to 2021, the UK reported seven MPOX cases, four linked to travel to regions where the disease is prevalent [10,11,12]. In July 2021, two people returning from Nigeria were diagnosed with MPOX in Texas and Maryland, USA [2,13,14]. The rapid global spread of the virus, primarily fueled by international travel and mobility, led to the World Health Organization (WHO) designating MPOX (MPXV) as a public health emergency of international concern (PHEIC) under the International Health Regulations [15]. On August 14, 2024, the WHO Director-General officially declared the MPOX outbreak a public health emergency, highlighting the rapid spread of the disease, the growing number of cases globally, and the urgent need for coordinated international efforts to control its transmission [16].

Over the years, vaccines have proven effective in infectious disease control and prevention, and MPOX is no exception [17,18]. Given the close relationship between the two viruses, existing vaccines, such as the smallpox vaccine, have shown efficacy in preventing MPOX infections [9,18]. However, the global response to the MPOX outbreak has exposed deep-seated challenges in distribution and equitable vaccine access [18,19]. Factors such as vaccine hoarding, a lack of manufacturing capacity in LMICs, logistical barriers, and intellectual property issues hinder equitable distribution [17,18]. This research examines the global challenges in distributing and ensuring equitable access to MPOX vaccines. In doing so, it seeks to contribute to ongoing efforts to achieve equitable health outcomes in the global fight against emerging infectious diseases.

## 2. Monkeypox Vaccines

Immunological reactions to a single Orthopox virus can recognize other Orthopox viruses and provide variable degrees of protection based on the similarity between the different Orthopox viruses. The viruses variola, vaccinia, and monkeypox (pox) are classified under the genus of Orthopox viruses. Because of shared antigens, vaccination can effectively prevent both variola and monkeypox. Recent research has demonstrated that the vaccinia vaccine, previously known for its efficacy in preventing smallpox, can also protect against the MPOX virus [5,20,21,22,23,24].

This immunological cross-reactivity has facilitated the development of several animal models of smallpox infection, which have been utilized to evaluate the effectiveness of vaccinations and antivirals [25]. Two factors are primarily responsible for this cross-reactivity. Firstly, the Orthopox viruses exhibit a significant level of sequence similarity, particularly in immunologically essential proteins, resulting in a substantial number of common immune epitopes [26,27]. The reaction includes a broad range of antibodies targeting a minimum of 24 membrane and structural proteins [28,29,30]. Similarly, T cell responses detect epitopes in a wide range of viral proteins, with CD4 T cells showing a preference for recognizing structural proteins [31], while CD8 T cells specifically target proteins that are encoded early in the viral life cycle, such as virulence factors [32,33]. A neutralizing antibody was established to protect against smallpox (produced by the variola virus) in humans and against other Orthopox viruses in animal models. Though not essential for protection, T cells act by contributing to viral clearance [34]. Some of the first evidence that vaccinia-specific immune responses can protect against MPOX came from investigations conducted in the 1980s. Three studies were conducted on chimpanzees, rhesus macaques, and cynomolgus macaques to determine the efficacy of smallpox vaccines against the MPOX virus using the Dryvax vaccine (first-generation smallpox vaccine). The result showed that almost all vaccinated animals were protected against the virus. The one exception to the above results was determined to have not responded to the vaccine. Although these studies were small, they suggested that smallpox vaccines offered some protection from the MPOX virus. This laid the groundwork for understanding the cross-protective immunity between smallpox and MPOX and has had a massive role in vaccine development and public health strategies [35]. The relationship between the rise in MPOX cases following the end of smallpox vaccination and a growing population with limited immune exposure has been hypothesized and associated with a growing population with limited immune exposure. Due to concerns about bioterrorism and increasing outbreaks of the MPOX virus, the WHO has continued to license the smallpox vaccines [35,36]. Smallpox vaccines are classified according to generation.

First-generation smallpox vaccines contained live vaccine viruses and were administered through bifurcated needles. Historically, patients with a characteristic pustule were protected against the virus. These vaccines, which include Dryvax^®^, APSV^®^, Lancy–Vaxina^®^, and L-IVP^®^, were last produced in the early 1980s. Rare but potentially life-threatening complications, such as post-vaccinal encephalitis, progressive vaccinia, and myopericarditis, mainly occurred in immunocompromised patients [35,37,38]. Dryvax was developed by Wyeth Laboratories, consisting of the New York City Board of Health (NYCBH) strain of vaccinia and the Lister/Elstree and Ikeda vaccinia virus strains, which were lymph-derived and cultivated on the skin of animals [36]. The routine use of this vaccine ended in the United States in 1972, the license for Dryvax was revoked in 2008, and the vaccine remains insufficient [21]. The second-generation smallpox vaccines contained replication-competent viruses produced in a controlled laboratory setting. The focus was on eliciting a similar immune response to the first generation while curbing adverse side effects [35,37,38]. The third generation of smallpox vaccines concentrated on attenuated vaccine strains such as LC16m8, MVA, NYVAC, and dVVL to improve the safety profile [37].

The United States (US) has authorized second- and third-generation smallpox vaccines for MPOX vaccination. Cell culture methods were utilized for their development to enhance the safety traits of these vaccines. In 2007, the US Food and Drug Administration licensed ACAM2000. It is generated from a clone of Dryvax. Originating from Emergent Bio Solutions, this vaccine is specifically designed for active vaccination against smallpox, excluding MPOX disease, in those with a high risk for smallpox virus infection [39,40]. While the efficacy of ACAM2000 in preventing MPOX has not been definitively shown, several studies have verified a certain level of immunogenicity. Additionally, one research study demonstrated protection against this virus by employing its first-generation progenitor, Dryvax [41,42]. Although the United States has a large stock of ACAM2000, this vaccine exhibits more adverse events and contraindications than the third generation [39]. The adverse effects associated with this vaccine injection included pain and swelling at the injection site, fatigue, and muscular discomfort in about 50% of the recipients. Additionally, 20–40% of the recipients experienced lymphadenopathy and headache, while around 20% had fever. About 20% of the recipients reported joint pain, backache, gastrointestinal pain, or nausea. More severe adverse effects encompassed widespread vaccinia, eczema vaccinatum, progressive vaccinia, post-vaccinal encephalopathy or encephalitis, and mortality. This vaccine was contraindicated for individuals with severely compromised immune systems, pregnant women or lactating mothers, those with cardiac disease or cardiac risk factors, individuals with active eczema, and infants under 12 months of age [36,43,44,45]. Additional vaccines based on the vaccinia virus were created by subjecting the virus to repeated passage in a primary cell culture or eggs to attenuate it [36].

In 2019, the FDA approved JYNNEOS, a third-generation smallpox vaccine [40]. It is a product of the Modified Vaccinia Ankara (MVA) attenuated strain, created through more than 500 serial passages of the vaccinia virus Ankara strain in chick embryo fibroblast cells. It is purified using tangential flow filtration and supplied as a frozen liquid suspension [35,42]. The vaccine is administered subcutaneously in two doses, 28 days apart, usually in the upper arm. JYNNEOS can also be administered intradermally, though this method is not recommended for individuals with weakened immune systems or a history of keloid scarring and is not preferred as the initial dose for post-exposure vaccination [46]. After two doses, the immune response is comparable to that of ACAM2000 but with fewer side effects [34,47]. MVA-based vaccines do not produce the typical skin reaction. Still, common mild side effects include pain at the injection site (85% of recipients), redness, swelling, itching, and induration at the injection site (40–60%), as well as fatigue, muscle aches, headaches (20–40%), nausea (17%), and chills (10%). Fever is uncommon and observed in about 2% of recipients; cardiac events were reported in about 2%, with no cases of myopericarditis [34]. The vaccine approval for the protection from smallpox in adults came from the European Medicines Agency (Amsterdam, The Netherlands) (EMA) in 2013 under the trade name IMVANEX and under the trade name IMVAMUNE from the Public Health Agency of Canada (PHAC) in the same year. In 2020, the PHAC expanded IMVAMUNE’s use to include MPOX and other Orthopoxvirus infections, and in July 2022, the EMA similarly extended IMVANEX’s use to cover monkeypox and vaccinia-related diseases [48].

LC16m8 is another third-generation vaccine developed from the Lister strain used in first-generation vaccines. Through multiple tissue culture passages and selection for an attenuated phenotype, the LC16m8 strain was produced and lacked a full-length, functional B5 membrane protein [34]. The vaccine is produced in cell culture using rabbit kidney cells and has been licensed since 1975 for active smallpox vaccination in Japan. In August 2022, Japan expanded the vaccine’s use to include protection against MPOX. Due to their attenuated nature, these vaccines have a better safety profile and can be given to immunocompromised individuals. The effectiveness of these vaccines against MPOX is inferred from animal studies showing protection against the MPOX virus in non-human primates vaccinated with these vaccines, as well as clinical trials that have demonstrated their immunogenicity in humans [36].

These vaccines can be used pre-exposure to prevent infection and illness or mitigate infection and disease post-exposure. Pre-exposure vaccination is recommended for individuals at the highest risk, with second- or third-generation vaccines providing the most effective protection. Post-exposure vaccination is ideally administered within four days of exposure to prevent infection. However, it can still be given up to 14 days after exposure to lessen the severity of the disease. Second- or third-generation vaccines are also preferred for post-exposure vaccination. In July 2022, authorities in Montreal, QC, Canada, administered at least 3000 doses [34]. In all cases, healthcare professionals should know who may qualify for vaccination so they can consult national health authorities about accessing vaccines from national stockpiles. Smallpox vaccines are not available commercially or privately.

## 3. Challenges in Global Distribution of Monkeypox Vaccines

The World Health Organization (WHO) has drawn attention to global competition for Mpox vaccines, with 35 countries vying for access to the 16.4 million currently available doses [49]. There is a growing concern that low-income countries may encounter challenges securing the vaccine [50]. Most notably, many African countries where Mpox has been prevalent for decades are yet to receive a single dose of the vaccine. Wealthier nations have obtained most of the available doses, limiting options for other affected countries, particularly those in Africa [50,51].

AIDS, COVID-19, and the current monkeypox outbreak are evidence of inequities in global health strategy. It can be observed that global health strategies for disease burden control are only put in place when the Global North (developed countries) is affected [3]. The concentration of global vaccine production in the Global North, with numerous vaccine production facilities in the United States, Japan, and Europe, has led to a reliance on vaccine supply from these regions. This reliance creates significant inequity in global vaccine coverage and insufficient domestic production, particularly in low- and middle-income nations [52]. Despite the establishment of vaccine production facilities by foreign pharmaceutical industries across Africa [53], they still produce less than 1% of all vaccines used on the continent [54].

Additional challenges, such as the lack of qualified personnel, specialized vaccine development, and manufacturing infrastructure, inadequate training, ineffective awareness campaigns, and difficulties collecting and processing critical immunization data, further complicate the situation [50].

Furthermore, challenges such as an unstable electricity supply and inadequate cold chain facilities could significantly impede comprehensive vaccine manufacturing in Africa [50].

## 4. Barriers to Equitable Access to Monkeypox Vaccines

*Advance Purchase Agreements:* High-income countries have leveraged advance purchase agreements (APAs) to secure priority access to newly produced vaccines. These agreements allow wealthier nations to stockpile vaccines, often at the expense of countries that may be more affected by monkeypox. This practice not only limits the availability of vaccines for lower-income nations but also reinforces inequities, as those countries cannot compete financially in the vaccine market. A more equitable approach would involve manufacturers setting aside a portion of their vaccine supply for global distribution, ensuring that all countries have access regardless of their economic status [55].

*Vaccine Stockpiling Reluctance:* Nations with large stockpiles of vaccines are often hesitant to share these resources. This reluctance stems from concerns about maintaining sufficient supplies for their populations amid outbreaks. However, this hoarding behavior exacerbates global disparities in vaccine access, particularly affecting historically endemic countries that experience the highest case fatality rates. To address this issue, countries should commit to contributing a portion of their stockpiled vaccines to an international distribution network, aligning their actions with global health commitments, and fostering collective security against outbreaks [55].

*Lack of Organized Mechanisms for Low-Income Countries:* Many low-income nations lack the infrastructure and financial means to procure vaccines independently, relying instead on international aid and charity. This dependency can lead to inconsistent access and delays in vaccination efforts, ultimately allowing the disease to spread unchecked. Establishing a coordinated global procurement and distribution system could help bridge this gap. Like the successful Revolving Fund for Access to Vaccines in the Americas, such a system would provide a structured approach for low-income countries to secure necessary vaccines, ensuring they are not left behind during public health emergencies [55].

*Vaccine Nationalism:* During health crises, many countries prioritize securing vaccines for their populations, a practice often called vaccine nationalism. This trend was evident during past outbreaks, such as those of H1N1 and COVID-19, where wealthier nations quickly acquired the majority of available doses, leaving low-income countries with limited access. The current landscape for monkeypox is no different, as high-income countries have disproportionately secured vaccine supplies, creating a significant gap in availability for those most affected. To combat this, international frameworks and agreements are needed to promote fair distribution practices and discourage hoarding [55,56].

*Inequitable Distribution of Existing Resources:* Even when vaccines are available, their distribution is often inequitable. Many countries with a high monkeypox prevalence receive little to no doses, while nations with minimal cases secure large quantities. This disconnect between the disease burden and vaccine access is unjust and highlights a critical need for allocation strategies prioritizing regions most in need. Developing criteria based on case severity and potential risk factors can help guide equitable distribution [55].

*Insufficient Surveillance and Reporting Mechanisms:* Countries needing more robust health surveillance systems often need help accurately documenting monkeypox cases. This limitation can lead to underreporting, resulting in these countries receiving fewer vaccines due to perceived lower needs. To address this, international support is crucial for building the surveillance capacity in vulnerable nations. Enhanced data collection and reporting will improve vaccine allocation decisions and help contain future outbreaks effectively [55].

## 5. Strategies to Improve Global Distribution and Access

The global distribution of the monkeypox vaccine has faced many challenges, especially in ensuring equitable distribution and access for vulnerable populations. It can be improved through (1) enhancing global health policies; (2) establishing a new public health structure [57]; (3) global health financing [58]; (4) addressing pharmaceutical monopolies [57]; (5) price transparency [59]; (6) enhancing vaccine confidence and uptake; (7) utilizing social media for communication [59]; (8) implementing ring vaccination strategies [60]; and (9) international cooperation and solidarity [61].

## 6. Recommendations

### 6.1. Strengthening Global Vaccine Manufacturing Capacity

Significant disparities in the fair access to necessary medical countermeasures, such as vaccines, were brought to light by the COVID-19 pandemic. The manufacturing capacity for pandemic vaccines is concentrated in too few countries. Eight of every ten vaccine doses produced have gone to high-income countries [61]. Expanding the manufacturing capacity across different regions is crucial to overcoming supply constraints, particularly in low- and middle-income countries (LMICs). This could involve transferring technology to these regions, fostering partnerships between pharmaceutical companies and local producers, and encouraging cooperation between players from the public sector, business sector, academic institutions, and civil society.

### 6.2. Targeting Countries with Elevated Risk Profiles and Limited Access to Resources

Nigeria, Ghana, and Cameroon have recorded about 15% of global monkeypox fatalities since 2022 but have fewer than 1% of the total cases worldwide [4]. Countries should be prioritized based on ethical principles. This prioritization could specify the order in which countries receive required doses or how large a share they receive from a given tranche. Under either method, historically endemic countries should receive the highest priority for monkeypox vaccines. After that, the priority should be determined using countries’ projected cases and severity of outcomes [62]. Vaccine distribution should prioritize regions experiencing the highest monkeypox incidence and populations at higher risk.

### 6.3. Overcoming Vaccine Nationalism Through Platforms Like COVAX

Platforms such as COVAX (COVID-19 Vaccines Global Access) could be adapted to include monkeypox vaccines, ensuring a more coordinated global response. Establishing an international task force similar to the ACT Accelerator (Access to COVID-19 Tools), used during COVID-19 could facilitate the distribution of monkeypox vaccines to the most vulnerable populations. COVAX was designed as an end-to-end coordination mechanism encompassing research and development and manufacturing, policy guidance, vaccine portfolio development, regulatory systems, supply allocation and country readiness assessments, transport logistics, vaccine storage and administration, and the monitoring of country coverage and absorption rates. It significantly alleviated the suffering caused by COVID-19 in the Global South (low-income countries). Today, the initiative has supplied 74% of all COVID-19 vaccine doses provided to low-income countries (LICs) during the pandemic. A total of 52 of the 92 AMC-eligible economies relied on COVAX for over half of their COVID-19 vaccine supply [63].

### 6.4. Strengthening Health Surveillance Systems

A significant contributor to vaccine access inequities, especially in low- and middle-income countries, is the underreporting of Mpox cases, especially in the central and west African regions [57]. This is mainly attributable to the weak health surveillance systems in rural areas, which prevent the early detection of cases, leading to the underreporting of cases. Due to this, the disease’s hotspots may not be readily identified early, consequently preventing vaccines from being allocated to such areas. To address this issue, we recommend strengthening health surveillance systems in these regions by providing early warning systems incorporating robust data collection and sharing mechanisms. This will help prompt the detection and identification of cases in this region. In addition, assisting financially and technically to the end of developing disease monitoring and reporting systems, especially in low- and middle-income countries, will help improve vaccine access and equity [57].

### 6.5. Addressing Vaccine Hesitancy

A key challenge in the equitable distribution of Mpox vaccines is the unwillingness of people to receive these vaccines despite their availability. This is especially common in the Global South due to a need for more awareness, misinformation, or both [50]. The addressing of vaccine hesitancy by healthcare workers, community leaders, and local influencers is paramount. This can be achieved via well-planned public health campaigns and community engagement aimed at adequately informing people about and sensitizing people to the importance of vaccines, debunking myths about vaccines, and encouraging at-risk populations to get vaccinated. This would increase vaccination and the acceptance of vaccines [64,65].

## 7. Conclusions

As the world continues to grapple with the Mpox epidemic, with some countries being more significantly affected than others, there is an urgent need for a collaborative and progressive approach towards ensuring that countermeasures are not only provided but also fairly and equitably distributed to affected countries. Overcoming the challenges facing global distribution and equitable access to the MPox vaccine through measures such as strengthening the global vaccine manufacturing capacity, overcoming vaccine nationalism through platforms like COVAX, and addressing vaccine hesitancy are crucial steps in the fight against the Mpox epidemic. The development of international research and development centers, as well as vaccine distribution centers, can improve vaccine availability, reduce scarcity, and address the need for vaccine nationalism or hoarding. A global perspective policy on the equitable distribution of vaccines not just for Mpox but for other diseases is crucial.

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
