# Peer review of "Challenges in Global Distribution and Equitable Access to Monkeypox Vaccines"

_viruses, 2024, doi:10.3390/v16121815_

Round 1

Reviewer 1 Report

Comments and Suggestions for Authors

In front of the unacceptable inequity in access to mpox vaccine and the global risk for a pandemics, this article is highly timely and reinforces neglected global health messages such recent WHO DG declaration.

The references are  well update with the most recent information.

Funding and logistical challenges are well considered. it seems to me that regulatory challenges at all levels( international but also national ) should be highlighted. But this should not delay the ongoing publication process.

Funding also seems to remain anle  key obstacle because of the unit high cost.If transfer of technologies for production and procurement of vaccines to low and middle countries would be provided, this would reduce the reliance of most affected countries  in terms of timely access.

Last point, maybe to be pointed out in the recommandation, it is the importance of the policy making development, including advocacy and commitment of political leaders over the world ( not only from Africa)

Author Response

Reviewers comment:

The reviewers indicate that your manuscript may require some improvement to
English. Please review the English and decide whether revisions are
Necessary. Last point, maybe to be pointed out in the recommandation, it is the importance of the policy-making development, including advocacy and commitment of political leaders over the world ( not only from Africa)

Response:

Thank you for your time and comment. We have taken note of this and made the corrections where necessary. 

Reviewer 2 Report

Comments and Suggestions for Authors

After evaluating the manuscript, I find it addresses an important topic in global health—equitable access to monkeypox vaccines and broader challenges in managing zoonotic outbreaks. However, while authors outline some key barriers, the manuscript lacks in both scientific novelty and actionable solutions that would meaningfully advance the field. Below, I provide a detailed review and suggest specific recommendations that could improve the manuscript’s rigor and overall impact.

Major Concerns

  1. Limited Scientific Novelty:
    • The manuscript repeats well-known barriers to vaccine distribution, such as vaccine nationalism, limitations in production, and logistical challenges. These issues were already broadly discussed in context of COVID-19 pandemic, and here the manuscript does not offer new data, perspectives, or frameworks. For this manuscript to contribute effectively to the field, it would need to introduce innovative strategies or propose unique approaches that are specific to monkeypox epidemiology and social-political context.

Recommendation: Adding any concept, as described in the similar article, the One Health approach might improve depth and relevance of the manuscript. Considering monkeypox's zoonotic nature, any concpet perspective could add understanding to transmission control strategies, which go beyond vaccination efforts alone (DOI: 10.7759/cureus.61867).

  1. Lack of Actionable Solutions:
    • While authors identify several broad challenges, the manuscript lacks concrete recommendations for public health authorities or policymakers. Suggestions like “strengthening manufacturing capacity” or “overcoming vaccine nationalism” are too broad and would benefit from more detailed, feasible steps.

Recommendation: Add focus on localized community engagement and education programs. Culturally relevant public awareness campaigns in affected regions can raise awareness and help address vaccine hesitancy. Other studies have emphasized this as critical for addressing misinformation and improving vaccine access (DOI: 10.7759/cureus.61867).

  1. Absence of Empirical Data:
    • The manuscript heavily relies on existing literature and theoretical discussions without presenting any new data. This lack of original data gives an impression of a review paper rather than an original research work. Including survey results on vaccine hesitancy, logistical barriers, or health outcomes in monkeypox-endemic areas could add practical value.

Recommendation: Incorporate empirical data where feasible, for example by using case studies, surveys of healthcare workers, or affected communities. Such data would help strengthen the recommendations and make manuscript’s findings more applicable to real-world settings.

Specific Suggestions for Practical Application

  1. International R&D and Vaccine Distribution Center:
    • To address both vaccine distribution issues and gaps in scientific research, the authors might propose the new ideas, as an International Research and Development (R&D) Center for Zoonotic Diseases and Vaccine Development or something like this. This center could focus on both monkeypox and other zoonotic threats, providing:
      • Collaborative Research: Uniting immunology, virology, and public health experts to study transmission dynamics, immunogenicity, and vaccine efficacy of monkeypox across diverse populations.
      • Testing and Distribution Protocols: Developing standards for testing vaccine effectiveness and ensuring safe distribution in high-risk and underserved areas.
      • Capacity Building: Training local healthcare professionals in outbreak response, diagnostics, and vaccination, empowering communities with more independence in handling outbreaks.

Establishing such a center would help to create a globally coordinated response, aiming at both research and swift outbreak control.

  1. Integrated Health Screening and Vaccination for Migrants and Travelers:
    • Considering that human mobility accelerates zoonotic disease spread, the authors could suggest cross-border health screening and vaccination programs targeting high-risk groups such as migrants and travelers. This approach may include:
      • Vaccination Access: Providing vaccines to individuals moving between endemic regions and reducing transmission risks.
      • Education Initiatives: Offering health education specifically designed for transient populations to explain monkeypox transmission and benefits of vaccination.
      • Data Collection: Implementing health screening programs among migrant populations to gather data on disease trends and immunity levels, informing future public health responses.

This approach would provide a practical strategy for reducing disease spread and simultaneously collecting valuable surveillance data to aid in future interventions.

Minor concerns

  1. Expanded Limitations Section:

While limitations are briefly mentioned, a more transparent discussion about the lack of empirical data and specific methodological constraints would improve rigor. Such limitations section would allow readers to more critically assess the findings.

Comments on the Quality of English Language
    • While mostly clear, the manuscript would benefit from more precise language. Some grammatical issues and phrasing errors could be improved with editing, and restructuring certain sections with clear subheadings would also enhance readability.

Author Response

Reviewers comment

  1. Limited Scientific Novelty:
    • The manuscript repeats well-known barriers to vaccine distribution, such as vaccine nationalism, limitations in production, and logistical challenges. These issues were already broadly discussed in the context of the COVID-19 pandemic, and here, the manuscript does not offer new data, perspectives, or frameworks. For this manuscript to contribute effectively to the field, it would need to introduce innovative strategies or propose unique approaches that are specific to monkeypox epidemiology and social-political context.

Recommendation: Adding any concept, as described in a similar article, the One Health approach might improve the depth and relevance of the manuscript. Considering monkeypox's zoonotic nature, any concept perspective could add understanding to transmission control strategies, which go beyond vaccination efforts alone (DOI: 10.7759/cureus.61867).

Response: Thank you so much; sadly, we couldn’t find a concept specific to Monkeypox. While the Paper recommended is insightful, it doesn’t exactly answer what we set out to achieve with this paper.

  1. Lack of Actionable Solutions:
    • While the authors identify several broad challenges, the manuscript lacks concrete recommendations for public health authorities or policymakers. Suggestions like “strengthening manufacturing capacity” or “overcoming vaccine nationalism” are too broad and would benefit from more detailed, feasible steps.

Recommendation: Add a focus on localized community engagement and education programs. Culturally relevant public awareness campaigns in affected regions can raise awareness and help address vaccine hesitancy. Other studies have emphasized this as critical for addressing misinformation and improving vaccine access (DOI: 10.7759/cureus.61867).

Response: We value this insight. The reason for the broad suggestions is that we are looking at the global space and not limiting it to a specific region. We are also looking at challenges in global distribution and equity for the MPX vaccine, hence our reason for not focusing much on community engagement and education programs. We have also added some necessary contributions from the suggested paper.

  1. Absence of Empirical Data:
    • The manuscript heavily relies on existing literature and theoretical discussions without presenting any new data. This lack of original data gives the impression of a review paper rather than an original research work. Including survey results on vaccine hesitancy, logistical barriers, or health outcomes in monkeypox-endemic areas could add practical value.

Recommendation: Incorporate empirical data where feasible, for example, by using case studies, surveys of healthcare workers, or affected communities. Such data would help strengthen the recommendations and make the manuscript’s findings more applicable to real-world settings.

Response: The paper was submitted as a Review paper with the intention of being a review commentary.

Specific Suggestions for Practical Application

  1. International R&D and Vaccine Distribution Center:
    • To address both vaccine distribution issues and gaps in scientific research, the authors might propose new ideas, such as an International Research and Development (R&D) Center for Zoonotic Diseases and Vaccine Development or something like this. This center could focus on both monkeypox and other zoonotic threats, providing:
      • Collaborative Research: Uniting immunology, virology, and public health experts to study transmission dynamics, immunogenicity, and vaccine efficacy of monkeypox across diverse populations.
      • Testing and Distribution Protocols: Developing standards for testing vaccine effectiveness and ensuring safe distribution in high-risk and underserved areas.
      • Capacity Building: Training local healthcare professionals in outbreak response, diagnostics, and vaccination, empowering communities with more independence in handling outbreaks.

Establishing such a center would help to create a globally coordinated response, aiming at both research and swift outbreak control.

Response: Thank you for this. We will include this suggestion in our recommendation.

  1. Integrated Health Screening and Vaccination for Migrants and Travelers:
    • Considering that human mobility accelerates zoonotic disease spread, the authors could suggest cross-border health screening and vaccination programs targeting high-risk groups such as migrants and travelers. This approach may include:
      • Vaccination Access: Providing vaccines to individuals moving between endemic regions and reducing transmission risks.
      • Education Initiatives: Offering health education specifically designed for transient populations to explain monkeypox transmission and benefits of vaccination.
      • Data Collection: Implementing health screening programs among migrant populations to gather data on disease trends and immunity levels, informing future public health responses.

This approach would provide a practical strategy for reducing disease spread and simultaneously collecting valuable surveillance data to aid in future interventions.

Minor concerns

  1. Expanded Limitations Section:

While limitations are briefly mentioned, a more transparent discussion about the lack of empirical data and specific methodological constraints would improve rigor. Such limitations section would allow readers to more critically assess the findings.

Response: Thank you very much.

Reviewer 3 Report

Comments and Suggestions for Authors

In the manuscript from Danladi and colleagues, the authors provide commentary on the global and equitable distribution of vaccines against the monkeypox virus (MPOX) due to more recent outbreaks in traditional endemic areas such as central and west Africa, but also in countries of Europe and America.  The authors point out that between 2022 and 2024, over 100,000 confirmed cases have occurred with 222 fatalities. The authors discuss the different vaccines for MPOX, global challenges in the distribution of vaccines, and barriers to equitable access to MPOX vaccines, and end the manuscript with recommendations.  Overall, I felt the manuscript covers an important topic in human virus diseases and was well-written except for the numerous typos discussed below.

Major comments:

1. Line 26: Challenges and Global should not be capitalized.

2. Throughout the manuscript, the references cited were in different formats. For example, references were either:

Sentence. (1,2).

Sentence  (1,2)

Sentence. (1,2)

Sentence (1, 2) next sentence

I believe that the format for cited references for this journal is ….sentence [1,2].

This needs to be corrected.

3. Line 39-40: “smallpox, a disease eradicated over three decades ago.”  The World Health Assembly declared smallpox eradicated on May 8, 1980, over 44 years ago. Please correct.

4. Line 92: Please provide a reference for the statement on this line.

5. Line 196-197: “….inequities in Global Health strategy…”  should be “inequities in global health strategy.”

6. Line 197-198: The sentence: “ It can be observed that Global health strategies for disease  burden control are only put in place when the Global North is affected.” should be re-written as: “It can be observed that global health strategies for disease burden  control are only put in place when the global north is affected.” Also, please define “global north.”

7. Lines 262-266: The statement, “Enhancing Global Health Policies, Establishing a New Public Health Structure, (58)Global Health Financing,(59) Addressing Pharmaceutical Monopolies, (58) Price Transparency, (60) Enhancing vaccine confidence and uptake, Utilizing social media for communication, (60)  Implementing ring vaccination strategies, (61) and International Cooperation and Solidarity(16)”  should be connected to the first part of the sentence (It can be improved through…..).  Additionally, there is a profuse use of capital letters.

Try this: “It can be improved through 1) enhancing global health policies; 2) establishing a new public health Structure (58); 3) global health financing (59); 4) addressing pharmaceutical monopolies (58); 5) price transparency (60); 6) enhancing vaccine confidence and uptake; 7) utilizing social media for communication (60); (8) implementing ring vaccination strategies (61); and 9) international cooperation and solidarity (16).”  

8) Line 290: Please define “ACT.”

9) Line 297: Please define global south. Also, Global South probably shouldn’t be capitalized.

10) If possible (and a big if), having a table of those countries and quantities with MPOX vaccines would be nice.

Author Response

Major comments:

  1. Line 26: Challenges and Global should not be capitalised.

Response: Thank you. We have addressed it.

  1. Throughout the manuscript, the references cited were in different formats. For example, references were either:

Sentence. (1,2).

Sentence  (1,2)

Sentence. (1,2)

Sentence (1, 2) next sentence

I believe that the format for cited references for this journal is ….sentence [1,2].

This needs to be corrected.

 Response: Thank you. We have addressed it.

  1. Line 39-40: “Smallpox, a disease eradicated over three decades ago.”  The World Health Assembly declared smallpox eradicated on May 8, 1980, over 44 years ago. Please correct.

Response: Thank you so much for this correction. It has been rectified.

  1. Line 92: Please provide a reference for the statement on this line.

Response: Thank you. It has been addressed.

  1. Line 196-197: “….inequities in Global Health strategy…”  should be “inequities in global health strategy.”

Response: Thank you. It has been addressed.

  1. Line 197-198: The sentence: “ It can be observed that Global health strategies for disease  burden control are only put in place when the Global North is affected.” should be re-written as: “It can be observed that global health strategies for disease burden  control are only put in place when the global north is affected.” Also, please define “global north.”

Response: Thank you. It has been addressed.

  1. Lines 262-266: The statement, “Enhancing Global Health Policies, Establishing a New Public Health Structure, (58)Global Health Financing,(59) Addressing Pharmaceutical Monopolies, (58) Price Transparency, (60) Enhancing vaccine confidence and uptake, Utilizing social media for communication, (60)  Implementing ring vaccination strategies, (61) and International Cooperation and Solidarity(16)”  should be connected to the first part of the sentence (It can be improved through…..).  Additionally, there is a profuse use of capital letters.

Try this: “It can be improved through 1) enhancing global health policies; 2) establishing a new public health Structure (58); 3) global health financing (59); 4) addressing pharmaceutical monopolies (58); 5) price transparency (60); 6) enhancing vaccine confidence and uptake; 7) utilising social media for communication (60); (8) implementing ring vaccination strategies (61); and 9) international cooperation and solidarity (16).”  

Response: Thank you. It has been addressed.

8) Line 290: Please define “ACT.”

Response: Thank you. It has been addressed.

9) Line 297: Please define global south. Also, Global South probably shouldn’t be capitalised.

Response: Thank you. It has been addressed.

10) If possible (and a big if), having a table of those countries and quantities with MPOX vaccines would be nice.

Response: Thank you for this. It would have greatly improved this paper; sadly, we could not find the data.  

Reviewer 4 Report

Comments and Suggestions for Authors

The article titled: Challenges in Global Distribution and Equitable Access to Monkeypox Vaccines describes the development and use of different vaccines against mpox infections and analyses the reasons of the unequal distribution and stockpiling of the vaccine by high-income countries, disregarding the need of the mpox vaccine in the countries with higher prevalence of the disease, particularly in Africa.

According to the WHO report (2024 Oct. 13.) sixteen countries on the African continent have reported mpox cases in the last six weeks. The most affected country in 2024 continues to be the Democratic Republic of the Congo (6169 confirmed cases, 25 deaths), followed by Burundi (987 confirmed cases, no deaths) and Nigeria (94 confirmed cases, no deaths).  https://www.who.int/publications/m/item/multi-country-outbreak-of-mpox--external-situation-report-40--13-october-2024#:~:text=Sixteen%20countries%20on%20the%20African,confirmed%20cases%2C%20no%20deaths).

Outside Africa, the highest number of confirmed cases in September 2024 was reported by Australia. The country is currently experiencing an increasing outbreak of clade IIb MPXV, affecting mainly men who have sex with men and are infected through sexual contact. https://reliefweb.int/report/democratic-republic-congo/multi-country-outbreak-mpox-external-situation-report-41-26-october-2024

According to the WHO, more than 90,000 cases of mpox have been reported since the 2022 worldwide outbreak, which resulted in 167 deaths, while a new outbreak in Africa since 2023 has resulted in over 18,000 cases and 617 deaths. Protopapas K, et al. Mpox and Lessons Learned in the Light of the Recent Outbreak: A Narrative Review. Viruses. 2024 Oct 16;16(10):1620. doi: 10.3390/v16101620.

The authors correctly indicate, that there are several reasons of the unequal distribution of the mpox vaccines, including the lack of appropriate surveillance in identifying the risk-groups, the lack of qualified personnel, specialized vaccine development and manufacturing infrastructure, inadequate training, ineffective awareness campaigns, and difficulties collecting and processing critical immunization data, etc. Obviously, it wouldn’t make any sense to start distributing the existing 16 million vaccines in any region without having the slightest information about the magnitude and location of outbreaks, the risk groups, etc. For providing and allocating the necessary vaccines to the countries in need would be the duty of international organizations, similarly to the COVID vaccines.

I recommend the publication of the article in the present form with minor correction of inserting some WHO data on the updated situation of mpox.

Author Response

The article titled: Challenges in Global Distribution and Equitable Access to Monkeypox Vaccines describes the development and use of different vaccines against mpox infections and analyses the reasons of the unequal distribution and stockpiling of the vaccine by high-income countries, disregarding the need of the mpox vaccine in the countries with higher prevalence of the disease, particularly in Africa.

According to the WHO report (2024 Oct. 13.) sixteen countries on the African continent have reported mpox cases in the last six weeks. The most affected country in 2024 continues to be the Democratic Republic of the Congo (6169 confirmed cases, 25 deaths), followed by Burundi (987 confirmed cases, no deaths) and Nigeria (94 confirmed cases, no deaths).  https://www.who.int/publications/m/item/multi-country-outbreak-of-mpox--external-situation-report-40--13-october-2024#:~:text=Sixteen%20countries%20on%20the%20African,confirmed%20cases%2C%20no%20deaths).

Outside Africa, the highest number of confirmed cases in September 2024 was reported by Australia. The country is currently experiencing an increasing outbreak of clade IIb MPXV, affecting mainly men who have sex with men and are infected through sexual contact. https://reliefweb.int/report/democratic-republic-congo/multi-country-outbreak-mpox-external-situation-report-41-26-october-2024

According to the WHO, more than 90,000 cases of mpox have been reported since the 2022 worldwide outbreak, which resulted in 167 deaths, while a new outbreak in Africa since 2023 has resulted in over 18,000 cases and 617 deaths. Protopapas K, et al. Mpox and Lessons Learned in the Light of the Recent Outbreak: A Narrative Review. Viruses. 2024 Oct 16;16(10):1620. doi: 10.3390/v16101620.

The authors correctly indicate, that there are several reasons of the unequal distribution of the mpox vaccines, including the lack of appropriate surveillance in identifying the risk-groups, the lack of qualified personnel, specialized vaccine development and manufacturing infrastructure, inadequate training, ineffective awareness campaigns, and difficulties collecting and processing critical immunization data, etc. Obviously, it wouldn’t make any sense to start distributing the existing 16 million vaccines in any region without having the slightest information about the magnitude and location of outbreaks, the risk groups, etc. For providing and allocating the necessary vaccines to the countries in need would be the duty of international organizations, similarly to the COVID vaccines.

I recommend the publication of the article in the present form with minor correction of inserting some WHO data on the updated situation of mpox.

Response: Thank you for this contribution. We submitted the paper before the publication of these data. We are curious if updating this will not affect our credibility and that of the paper as well, seeing that the submission and acceptance date will be behind when the publication date of these new data from WHO. We are open to still improving it if you advise that it wouldn’t affect it.

Kind Regards.

Round 2

Reviewer 2 Report

Comments and Suggestions for Authors

Dear authors,

After reviewing the changes, I find that the manuscript still does not meet publication standards.

First, scientific novelty is still lacking. While your work reviews well-known barriers like vaccine nationalism and logistical challenges, it would benefit significantly from novel perspectives or strategies unique to monkeypox. 

The manuscript also lacks actionable solutions that could be realistically implemented by policymakers or public health authorities. A global perspective is essential, yet region-specific, culturally relevant strategies for community engagement and vaccine education in affected areas are equally critical. Such targeted recommendations could deepen the manuscript’s impact but are absent in the current version.

Additionally, the manuscript would be strengthened by empirical data. Although presented as a review, including empirical elements—such as case studies, surveys, or specific data on vaccine hesitancy in monkeypox-endemic areas—would add depth and make the findings more actionable. Without this, the work risks appearing as a reiteration of established literature rather than a new contribution to the field.

Finally, while the manuscript includes high-level recommendations, practical application details are missing

For these reasons, I must recommend a rejection at this stage.